# Molecular and Cellular Complexity of Glioma. Focus on Tumour Microenvironment and the Use of Molecular and Imaging Biomarkers to Overcome Treatment Resistance

**DOI:** 10.3390/ijms21165631

**Published:** 2020-08-06

**Authors:** Silvia Valtorta, Daniela Salvatore, Paolo Rainone, Sara Belloli, Gloria Bertoli, Rosa Maria Moresco

**Affiliations:** 1Department of Medicine and Surgery and Tecnomed Foundation, University of Milano—Bicocca, 20900 Monza, Italy; silvia.valtorta@unimib.it (S.V.); daniela.salvatore@unimib.it (D.S.); paolo.rainone@unimib.it (P.R.); 2Nuclear Medicine Department, San Raffaele Scientific Institute (IRCCS), 20132 Milan, Italy; sara.belloli@ibfm.cnr.it; 3Institute of Molecular Bioimaging and Physiology (IBFM), CNR, 20090 Segrate, Italy

**Keywords:** glioma, cell heterogeneity, tumor-microenvironment, miRNA, therapy resistance, molecular imaging, PET, predictive biomarkers, prognostic biomarkers

## Abstract

This review highlights the importance and the complexity of tumour biology and microenvironment in the progression and therapy resistance of glioma. Specific gene mutations, the possible functions of several non-coding microRNAs and the intra-tumour and inter-tumour heterogeneity of cell types contribute to limit the efficacy of the actual therapeutic options. In this scenario, identification of molecular biomarkers of response and the use of multimodal in vivo imaging and in particular the Positron Emission Tomography (PET) based molecular approach, can help identifying glioma features and the modifications occurring during therapy at a regional level. Indeed, a better understanding of tumor heterogeneity and the development of diagnostic procedures can favor the identification of a cluster of patients for personalized medicine in order to improve the survival and their quality of life.

## 1. Background

Glioma is the most diffused Central Nervous System (CNS) tumour in adults that accounts for the 75% of all the primary brain cancers [1]. Glioblastoma (GBM, WHO classification IV grade glioma), covers up to 54% of all gliomas and 16% of the total primary brain tumours [2]. The standard therapy relies on the Stupp protocol, officially proposed in 2005 [3,4]. The Stupp protocol is based on tumour surgical resection, followed by postoperative radiotherapy (RT), 60 Gy/30 fractions as standard, and concomitant plus adjuvant Temozolomide (TMZ) treatment [5]. Despite an increase in overall survival, the prognosis remains poor because of the development of GBM resistance to TMZ [6]. Alternative treatments are under evaluation, but the high intra- and inter-tumour heterogeneity of GBM and its intracranial localisation make the development of new therapies for this tumour a big challenge [7]. An important role in lesion heterogeneity is played by the tumour microenvironment (TME), composed by different cell populations, as immune cells, fibroblasts, precursor cells, endothelial cells, signalling molecules, and extracellular matrix (ECM) components [8,9]. In recent years, the use of new epigenetic molecules, such as non-coding RNA and microRNAs (miRNAs), has emerged in GBM characterization [10]. miRNAs are small non-coding RNA, able to modulate the level of expression of their mRNA targets at the post-transcriptional level, influencing in this way all cancer hallmark functions [11]. These molecules could be isolated in human tissues as well as in biofluids (saliva, urine, tears, cerebrospinal fluid, etc) and could represent tumor biomarkers thanks to their cell-specificity, expression, and stability [12,13]. In GBM, miRNAs represent potential biomarkers for the diagnosis, prognosis, and prediction of therapeutic response [14,15,16]. In addition, modulation of miRNA expression represents a suitable target for the development of new therapeutic tools [17]. The high heterogeneity and biochemical complexity of GBM make the development of preclinical models that recapitulate human disease a crucial issue [18]. In this context, the availability of non-invasive techniques that allow the longitudinal monitoring of tumour modifications during growth or under treatment is relevant for a better characterization and comparison of animal models representing patients [19]. Magnetic Resonance Imaging (MRI) represents state-of-the-art processes in the diagnostic management of glioma [20]. On the other hand, in vivo Positron Emission Tomography (PET) molecular imaging, associated with selected radiopharmaceuticals, allows the in vivo evaluation of specific biological features of the tumour and of its microenvironment at both clinical and preclinical levels [21]. In this review, molecular and cellular mechanisms involved in GBM resistance will be presented and discussed with the potential use of an in vivo multimodal imaging approach for the in vivo monitoring of morphological and molecular modifications in tumour cells and in the surrounding tissues and their association with response, resistance, or relapse to therapy.

## 2. Glioblastoma Cell Heterogeneity, Mutations, and Models

The morphology and biology of GBM present several challenges for treatment development, including infiltrative growth, pathological angiogenesis, the presence of necrotic regions that favour treatment resistance, and lesion relapse or recurrence [22,23,24]. One of the biggest issues is represented by inter and intra-tumour heterogeneity [25]. As described by Patel et al., a heterogeneous population of cells with different gene expression profiles rules GBM and this heterogeneity has been described also within the same lesion [26]. In addition, chemotherapy selects drug resistant cells that progressively replace the sensitive clones [27]. Recent advances in single-cell RNA-sequencing techniques have revealed the genomic landscape of GBM; indeed, GBM usually presents different genetic alterations that drive aberrant development programs and make GBM cells different between each other [28]. In particular, GBM is a hierarchically arranged tumour where cancer stem-like cells (CSCs) receive critical support signalling from their microenvironment [29]. The CSCs lodge in perivascular niches where the close adjacency to the endothelial vascular cells favours their stem cell-like state [30]. However, another stem-like tumour cell subpopulation resides in hypoxic regions far to the blood vessels, because of their capability to adapt to a low oxygen microenvironment, as recently reviewed by Najafi M et al. [31,32,33]. Tumour stem, non-stem, and normal cells incur in a communication stream to provide signals for the supporting of the cell state [34]. Differentiated progeny and blood vessels stimulate CSC maintenance through the production of cytokines, nitric oxide, Notch ligands, and the extracellular matrix [35,36,37,38]. CSCs are not passive receivers of microenvironmental stimuli, indeed CSCs support angiogenesis through angiogenic growth factor signalling, lead the differentiation of progeny, and regulate lineage plasticity toward vascular pericytes [39,40]. The CSC plasticity state is, therefore, influenced by the cellular microenvironment which couples autonomous and extrinsic cues [41]. In addition, GBM exhibits an exclusive tumour microenvironment that regulates immunity response, growth stimuli, invasion, and adaptation to hypoxia [42]. For these reasons, there is the need for a deeper comprehension for the genetic and epigenetic switches that promote GBM heterogeneity to better understand tumour development and the clinical outcome of specific patients’ subpopulations. Several studies that investigated GBM heterogeneity by single-cell RNA-sequencing strongly suggest that cellular heterogeneity in GBM is closely related to gene expression programs that are normally active during neurodevelopment [43,44,45]. Indeed, the GBM cell differentiation state often reflects a hybrid condition with similarity to various neuronal lineages [46]. An important question that arises from this is whether partial similarity to a neurodevelopmental cell line also implies that GBM cells preserve the phenotypes and functions of their healthy counterparts [47]. Bhaduri and colleagues have addressed this question investigating GBM heterogeneity by single-cell RNA-sequencing and dissecting its cellular composition. The authors observed that individual GBM tumours have heterogeneous cellular organization, with sub-populations of cells which partially recapitulate the expression programs of glial and neuronal lineages at various degrees of differentiation. Several of these cellular subpopulations also expressed recognized markers for glioma stem cells [48]. Their findings are focused on a cluster of GBM cells that present similarities to outer radial glia (oRG) cells, expressing the oRG marker PTPRZ1 [49]. The oRG cells are widely known to be neural stem cells involved in normal brain development, with an important migratory behaviour called mitotic somatic translocation (MST). The authors demonstrated that PTPRZ1 positive cells undergo MST and confirmed their role in the invasion of GBM cells, suggesting that this mechanism of normal oRG migration is active in a subset of GBM cells and may promote their invasion into brain parenchyma [50].

Previous studies, realised on the patients-derived tumour tissue, led by The Cancer Genome Atlas (TCGA) and the Repository for Molecular Brain Neoplasia Data (REMBRANDT) have grouped genomic, transcriptomic, and proteomic data which were annotated with clinical and, in some cases, MRI data [51,52]. These projects allowed glioblastomas to be grouped at the level of RNA expression. A recent report published in *Science* by Puchalski et al. introduced the Ivy Glioblastoma Atlas, a free accessible online database that correlates the molecular and anatomical features of GBM and provides a precious resource to investigate tumour heterogeneity [53]. Indeed, single cell RNA sequencing showed that glioblastoma subtype biomarkers are variably expressed at cellular levels within the same tumour. Importantly, this work also defined the potential prognostic implications of intra-tumour heterogeneity, adding an anatomical point of view to these approaches. The Atlas was based on 42 glioblastomas that had been resected from 41 patients. Histopathological assessment and RNA in situ hybridization were performed on the whole cross sections of the tumour samples, by laser microdissection, and were separated in five anatomical areas with differential profiles. The areas were classified into infiltrating tumour, cellular tumour, leading edge, pseudo-palisading cells around necrosis (PAN), and microvascular proliferation (MVP) [26]. The most representative prognostic marker to be investigated in the Atlas was the O6-methylguanine-DNA-methyl transferase (MGMT) promoter hypermethylation, which is overexpressed in the tumour tissue and is considered essential for the GBM cellular organisation network [54]. MGMT overexpression confers resistance to alkylating agents and its function is inhibited by MGMT promoter hypermethylation [55]. Nevertheless, several tumours show resistance to treatment despite the detectable hypermethylation of the MGMT promoter and one possible explanation resides in intratumor heterogeneity that limits the results obtained by tissue sampling [56]. Moreover, the detected pathways from the results of the Ivy Glioblastoma Atlas suggest that different pathways are involved, including that of neuronal development in the leading edge, precursors differentiation in the cellular tumour, the hypoxia inducible factor (HIF) system in PAN, and the tumour necrosis and HIF in MVP, which also displayed a hyperactivation of PI3K/AKT/mTOR signalling [53].

Theoretically, the tumour complexity and the low survival time after GBM diagnosis requires an opportune platform for tumour modelling and therapy testing. Nowadays, preclinical models include cell lines, organoids, xenografts, and genetically engineered mouse models [57]. Conventional GBM models regard either in vitro cell culture models, which do not recapitulate the heterogeneity of the primary tumour and lack the TME, or in vivo animal models that are logistically complex and limited by across species variations including genetically engineered models [58,59,60]. Patient-derived xenograft (PDX) models from human GBM are actually considered the most accurate strategy to study in vivo cancer cells’ environmental interactions. However, the use of PDX models is often hampered by donor availability, limited propagation, and lack of a functional immune system [61].

An emerging alternative, that may integrate the advantages of both of these classical approaches, is represented by the organoids generated from human samples. In 2016, Hubert et al. generated Organoid-derived Glioblastoma xenografts (GBOs), using patient-derived glioma stem cells (GSCs) and induced Pluripotent Stem Cells (iPSCs) to recapitulate tumour invasion, hypoxic gradients, and tumour stem cell heterogeneity using an updated procedure [62]. Afterwards, in two recent studies of Ogawa et al. and Linkous et al. human cerebral organoids were generated, adapting the culture method of Lancaster and Knoblich, and subsequently combining patient-derived GSCs with human embryonic stem cells (hESCs), in order to generate a cerebral organoid-derived glioblastoma model (GLICO). Although promising, these models lack the in vivo microenvironment, neuronal circuits, vascular circulation, and immune system [63,64,65]. To address this problem, Mansour et al. observed that only organoids developing vascularization can survive, following implantation into immunodeficient mice. To provide a vascularised and functional in vivo model of brain organoids, they established an efficient engraftment method using iPSCs cells. The authors demonstrated that the combination of human neural organoids and an in vivo physiological environment into the adult mouse brain may better reproduce the disease model [66]. In a recent study, Jacob et al., through single cell RNA sequencing and histopathology assays, confirmed the observations of Mansour et al. on in vivo GBO models, such as hypoxia gradients, vasculature, TME composition, mutations, and cell heterogeneity of parental tumours, generating a living GBO biobank thus making an attractive preclinical model to reproduce the lesion biology of patients [67].

### MicroRNA and GBM Microenvironment

MiRNAs are an endogenous class of about 20 nucleotides in length non-coding RNAs which are very abundant in the cells, with several regulatory functions. Indeed, miRNAs have a main role in the regulation of mRNA transcription by silencing or the degradation of target sequences. The complementarity of the mature miRNA sequence and the seed sequence (i.e., the sequence recognised by the miRNA on its putative target) determines the physiological role of the miRNA within the cell. Therefore, on the basis of the complementarity of the miRNA sequence with that of the seed sequence, the formation of a double strand allows the degradation of the target mRNA or its silencing due to the translation block [68,69]. The expression level of a selected miRNA could be easily determined by real time-PCR (RT-PCR) amplification, either on a tissue biopsy sample or on human biofluids [70,71]. Furthermore, in situ hybridization could be helpful in the localization of a specific miRNA inside the cells or tissues and in its quantification [72]. This feature allows miRNAs to be used either as diagnostic or prognostic biomarkers able to indicate the course of the pathology [73]. Moreover, the possibility of exogenous modulation on miRNAs makes them interesting molecules as a potential therapeutic target. Indeed, in vitro experiments on cell lines using synthetic oligonucleotides with a reverse and complement sequence of a miRNA could be used as an antisense sequence, to silence an endogenous molecule [74]. Vice versa, if a GBM has a decreased miRNA, the use of a synthetic sequence “mimicking” the silenced miRNA could re-establish its expression level, preventing the further development, growth, or metastasis of the GBM cells [17].

The main challenge in miRNA use is to identify the correct single or group of miRNAs with a primary role in GBM onset, development, or response to therapy. Bioinformatics in silico analyses help the biologists in the selection of miRNAs with a differential expression in GBM compared to normal tissues (diagnostic miRNAs) or those with a differential expression in different grade of the pathology, or those able to predict the prognosis of the patients (prognostic miRNAs) [75]. For instance, in 2013 Moller and colleagues proposed a miRNA signature of 256 upregulated miRNAs (notably miR-10b, miR-17-92 cluster, miR-21, and miR-93) and 95 downregulated miRNAs (including miR-7, miR-34a, miR-128, and miR-137) for the diagnosis of GBM [76,77]. Furthermore, individual miRNAs have been correlated with different glioma stages, such as in a work of Malzkorn and colleagues, where 12 upregulated miRNAs (miR-9, miR-15a, miR-16, miR-17, miR-19a, miR-20a, miR-21, miR-25, miR-28, miR-130b, miR-140, and miR-210) and two downregulated miRNAs (miR-184 and miR-328) were found to be associated to the spontaneous progression of GBM from the WHO grade II to the WHO grade IV [78].

Each miRNA could regulate more than a hundred targets and as a single mRNA could be the target of more than one miRNA, this creates a complicate regulatory network that impacts on several hallmark functions of different tumours, including GBM [79]. This regulatory network could possibly also affect the extracellular space, as several miRNAs could be secreted in the surrounding environment, including in cerebrospinal fluids, tears, saliva, or blood/serum [80]. This makes miRNAs important molecules for the intercellular or tissue communication, together with cytokines and chemokines [81]. The exact mechanism of miRNA secretion is not clear, but some hypotheses identify cells and apoptotic bodies as a source of secreted miRNAs; gap junctions, exosomes, extracellular vesicles, and tunnelling nanotubes were described in many tumours as possible passages for miRNAs from cell to cell [82].

Regarding the communication of a GBM cell with the surrounding environment, several miRNAs released from GBM are able to influence the angiogenesis. This process favours the tumour growth and potentially its invasion and metastasis. The increase in tumour mass can generate hypoxic areas, in which cellular metabolism also changes in relation to the surrounding microenvironment. Increased vascular endothelial growth factor (VEGF) release and other pro-angiogenic factor levels stimulate endothelial cell proliferation and the formation of leaky blood vessels, which are ineffective in supplying oxygen and chemotherapy, thus promoting hypoxia and the development of treatment resistance [83].

Several miRNAs have been described to have a role in angiogenesis. We summarized some of them in Table 1. For example, miR-613 was decreased in glioma cells and in vitro experiments revealed that its overexpression suppressed invasion, proliferation, and angiogenesis. Vascular endothelial growth factor A (VEGFA) was found as one of the targets of miR-613; in GBM the loss of miR-613 caused increased levels of VEGFA, which, in turn, led to increased angiogenesis [84].

On the contrary, miR-26a is amplified in GBM, where it has a main role in glioblastoma stem cell control: in this population, miR-26a is highly expressed and secreted by exosomes. These structures transport miR-26a into microvascular endothelial cells, where it silences PTEN expression, activating the PI3K-Akt pathway [85].

The miR-9 is also frequently up-regulated in glioma cells, where it enhances cell proliferation, migration, invasion, and new vessel formation, possibly by the inhibition of COL18A1, THBS2, PTCH1, and PHD3 and the promotion of the HIF-1α/VEGF signalling pathway [86]. Moreover, its target RGS5 plays a central role in vascular growth [87]. The silencing of RGS5 by siRNA treatment or by miR-9 overexpression, such as in GBM, stimulates endothelial growth [88]. The glioma microenvironment is mainly composed of microglia, astrocytes, and macrophages [89]. These cells represent the brain immune system displaying different homeostasis functions and regulating synaptic activity by the production of cytokines, chemokines, growth factors, and metabolites [90]. Few studies demonstrated the correlation between the number of microglia/macrophage cells and glioma, describing an activated state morphology particularly for the high-grade [91,92,93]. This population has been defined as glioma-associated microglia/macrophages (GAMMs) and could be influenced by the tumour microenvironment to support invasion [94].

Glioma cells produce IL-1, a potent inducer of angiogenesis and invasion, that in glial cells strongly upregulates miR-155 implicated in inflammation-mediated cancer development [95]. Other IL-1-induced miRNAs involved in inflammation are miR-21 and miR-146, upregulated in gliomas [96]. On the contrary, miR-146 is a negative regulator of astrocyte-mediated inflammation, migration, and invasion, and it decreases the expression of TRAF6 and MMP16 [97,98].

## 3. Chemoresistance and Treatments

### 3.1. Resistance to Temozolomide

Lack of response to TMZ may be present from the beginning of therapy or acquired during the treatment. The reasons are complex and only partially known, but a non-complete response favours tumour relapse and recurrence [114]. The MGMT promoter methylation status is a key parameter for counteracting TMZ cytotoxicity. Low MGMT promoter methylation entails a higher enzyme activity, a phenomenon which helps to acquire resistance to TMZ [115].

Cells lacking MGMT or which have defective Homologous Replication (HR) are responsive to TMZ treatment [116]. For this reason, MGMT is indicated as a biomarker for TMZ treatment response in a clinical setting [116,117,118,119].

Another mechanism by which TMZ affects DNA repair systems is mediated by N3-methyladenine and N7-methylguanine adducts, which are Base Excision Repair process (BER) targets. In fact, these residues induce Poly (ADP-ribose) polymerase (PARP) nuclear enzyme activation, which, by the PARylation (poly-ADP-ribose chain synthesis), recruit three components on the DSBs areas (XRCC1, pol-beta, and DNA ligase), by triggering the BER activation [120]. Since the PARP plays a key role in the BER mechanism, PARP inhibitors (PARPi) have been proposed and studied in combination to TMZ treatment for overcoming the currently defined resistance to TMZ [121].

Briefly, in a recent study of Higuchi et al., TMZ sensitivity has been restored by treating mismatch repair mechanism (MMR)-deficient resistant gliomas with PARPi by a BER-independent mechanism [120].

Beyond DNA, post-translational modifications on histones are also able to play a pivotal role in mediating gene expression, and this phenomenon might help to elucidate why MGMT is not always an efficient response biomarker to TMZ treatment. In fact, in a work of Chen et al. in 2018, an enhancer was described to activate MGMT gene expression, despite its highly methylated promoter. In detail, they have identified an enhancer sequence positioned between the promoters of MGMT and of Ki67 (a widely recognised marker of cell proliferation, also in GBM) which was able to activate not only MGMT expression but also Ki67 expression, by favouring GBM proliferation, both in vitro and in vivo. Moreover, the authors have also shown that the acetylation levels of the nucleosome H3K27, which borders the K-M enhancer, are capable of promoting MGMT expression in spite of its methylated promoter. They have observed the abrogation or the delay of resistance to TMZ treatment through the inhibition of the K-M enhancer [122].

Another important factor involved in resistance to TMZ is represented by the BER system activity, which counteracts cytotoxic effects of TMZ: when BER is abrogated, TMZ is able to exert its cytotoxic power [123]. This year, Belter et al. have shown that TMZ is able to totally augment the rates of the main epigenetic marker contained in tumour cells’ DNA, the 5-methylcytosine (m5C), in different GBM cell lines, by inducing a cellular hypermethylation state. During the reaction, S-adenosyl methionine (SAM) results in the unique Alkyl residues donor [124].

TMZ, beyond its alkylating effect, is able to induce other cellular mechanisms, such as the Endoplasmic Reticulum (ER) stress, Reactive Oxygen Species (ROS) production, and autophagy activation [125]. The tight relationship between the ER-stress and chemoresistance to TMZ has been already described, since TMZ toxicity also relies on promoting the chaperone protein folding role and on enhancing the degradation pathways, as the ER-mediated and the autophagic ones [126].

In a recent work, GBM MGMT-deficient cells with acquired resistance to TMZ treatment showed increased expression of dynein, cytoplasmic 2, heavy chain 1 (DHC2 or DYNC2H1), which is strictly related to TMZ-mediated cytoskeletal rearrangements [127].

In another recent study, the role of the bradykinin-bradykinin receptor B1 (BDKRB1) axis has been shown to modulate some GBM genes and pathways, as the Aquaporin-4 (AQP4/aqp4) expression and the calcium release within the cytosol. This process leads the mitogen-activated protein kinase (MAPK) pathway activation. The authors have demonstrated that the bradykinin/BDKRB1 increased the aqp4 expression through BDKRB1 mediated Ca2+ influx, which induces MEK4 phosphorylation and, consequently the downstream ERK1/2-NF-κB induction both in vitro and in vivo in human and murine cells. Overall, these data have suggested the key-role of bradykinin-BDKRB1-aqp4 action on cytoskeleton and morphological changes, as GBM oedema formation and its invasive and migratory power [128].

### 3.2. Pharmacological Strategies to Overcome GBM Resistance

Several therapeutic strategies have been developed in order to prolong patients’ overall survival (OS) and to overcome GBM resistance. Some of these therapies have completed clinical trials or are under clinical evaluation alone or in combination with TMZ; others are still confined to a pre-clinical scenario.

The angiogenic inhibitor Bevacizumab (a humanized monoclonal antibody targeting VEGF) has been approved by the Food and Drug Administration (FDA), but not by the European Medicines Agency (EMA), for patients with recurrent GBM. Despite the initial expectation based on the mechanism of action of the drug, no effect on OS, and only an increase of response rate in a subset of patients has been reported [129]. However, in a phase III clinical trial (EORTC 26101), the addition of Bevacizumab to Lomustine in 432 GBM patients previously treated, showed an increase in the time to disease progression or death compared to Lomustine alone (median progression-free survival = 4.2 vs. 1.5 months, HR = 0.52, 95% confidence interval = 0.41–0.64) [130].

Lomustine is a cross-linking agent able to produce amino acid carbonylation. In a recent study, Herrlinger et al. showed in a phase III randomized trial that Lomustine (CCNU) associated with RT/TMZ prolonged OS in comparison with an RT/TMZ standard regimen (median OS of 48.1 months versus 31.4 months, respectively) [131,132].

Lomustine has been recently evaluated in patients with recurrent GBM in combination with regorafenib (REGOMA), an inhibitor of angiogenic, stromal, and oncogenic receptor tyrosine kinase. Regorafenib plus Lomustine increased patients’ overall survival, being 7.4 months (95% CI 5.8–12.0) in the regorafenib group and 5.6 months (4.7–7.3) in the Lomustine group [133].

The block of hyperactivated Epidermal Growth Factor Receptors (EGFRs) represents another potential strategy. Specific interaction between Nimotuzumab, a humanized anti-EGFR antibody, and the constitutively activated EGFRvIII receptor, was able to inhibit EGFR signalling intracellular cascade, thus abrogating tumour proliferation, angiogenesis, and chemoresistance. The drug has been evaluated in different clinical trials; however, its effect is still controversial [134]. Recently, Du and colleagues evaluated in a phase II trial (NCT03388372) the efficacy of Nimotuzumab associated with TMZ/RT and concomitant TMZ in patients with newly diagnosed GBM carrying EGFRvIII [135]. The median OS and PFS were 24.5 and 11.9 months, respectively. These data may be of interest for a phase III clinical randomized controlled trial [136].

The PARP family of enzymes is an attractive target for the sensitization of GBM to chemotherapy because of their activity on DNA repair and metabolism. PARP inhibitors (PARPi) have been evaluated in different clinical trials, in association with TMZ, for treating newly diagnosed and recurrent glioblastoma patients [120]. In preclinical studies the association of TMZ and Veliparib showed promising results in GBM models [137,138]. In addition, Veliparib increased the TMZ cytotoxic effect in the PDX of newly diagnosed GBM with high MGMT promoter methylation [139]. However, in a phase II clinical study, Veliparib added to an RT/TMZ standard treatment in unmethylated MGMT patients with newly diagnosed GBM resulted in no ameliorated outcomes compared to the standard regimen [140]. A phase III clinical trial (NCT02152982) in which this drug is added to TMZ in treating MGMT-methylated, newly diagnosed GBM patients, is still ongoing [141]. Another PARPi, Olaparib, has been included in a phase I dose escalation study (OPARATIC trial) in combination to the standard therapy in recurrent GBM patients. The results of the study have been encouraging, with 45% of patients showing a PFS at six months. At the moment, researches are focused on the identification of potential markers of treatment efficacy [142].

An emerging strategy is represented by oncolytic viruses. Kiyokawa et al. reported the preclinical efficacy of the combination therapy of TMZ with the oncolytic adenovirus DNX-2401 (also named Delta-24-RGD) [143]. This drug and other oncolytic viruses are currently undergoing clinical evaluation alone or in combination with other drugs (check-point inhibitors, as, for instance, Pembrolizumab alone or in combination with the Heat Shock Protein Peptide Complex-96, HSPPC-96) in phaseI/II trials. Despite the efforts, the results obtained so far are not encouraging [144,145,146,147]. The metabolism and immune system represent emerging targets, but mainly under preclinical or early clinical investigation. Check-point inhibitors, IFN-β of human bone marrow derived-mesenchymal stem cells (MSC-IFN-β), the janus kinase (JAK) 1/2 inhibitor Momelotinib (MTB or even CYT387), or the biguanide Metformin, showed encouraging results when associated with TMZ or other molecules; however, clinical data, when present, are not definitive [148,149,150,151,152,153,154]. A list of the molecules cited above is reported in Table 2.

### 3.3. Role of miRNAs in GBM Chemoresistance

The current grading system, based on the histological classification, does not give any indication of the correct therapeutic approach for the patients [171]. The identification of drug sensitive patients with an affordable, reliable, and easy-to-use method could be helpful in addressing the patients to the right therapeutic option. In this view, miRNAs could be presented as indicative molecules for patient selection for a specific treatment. The choice of the right miRNA to be used as a drug-resistance associated molecule is still a challenge. The recent literature proposes several examples of miRNAs that could be used as potential biomarkers for the patients’ classification.

Looking for “microRNAs and glioblastoma and chemoresistance” publications of the last five years in PubMed, we found 30 papers. Two have been shelved because they speak of long non-coding RNA, and one was withdrawn by the authors before the publication. Twenty-seven manuscripts were analyzed to understand the role of miRNAs in the chemoresistance development (summarized in Table 3 and Figure 1). Most of the publications describe the role of single miRNA in this process, without proposing a miRNA signature associated with chemoresistance.

Several papers proposed a single miRNA with a role in the development of glioblastoma resistance to Temozolomide; one of the most frequent reported in the literature is miR-1238. This miRNA was found upregulated in TMZ-resistant glioblastoma cell lines and their exosomes, and in clinical tissues and sera of patients; its overexpression was able to modulate the CAV1/EGFR pathway. The presence of this miRNA in exosomes could influence the behaviour of the surrounding tissue, leading to the spreading of the resistant phenotype to surrounding tumour cells [172]. It has been already discussed that exosomal miRNAs allow the communication of the tumour cells to the neighbouring ones, as occurs with exosomal miR-151. Indeed, exosomal miR-151a enhances chemosensitivity to TMZ in drug-resistant GBM, possibly controlling XRCC4-mediated DNA repair. XRCC4 encodes for a vital protein of the double strand break repairing process: XRCC4 protein forms with DNA ligase IV a heterodimer that covalently joins the broken DNA ends to help cell survival. XRCC4 expression levels are found significantly downregulated in glioma tissue, indicating crucial roles of XRCC4 in brain carcinogenesis [173].

The exosomal communication among cells has been also been proven to increase the tumorigenicity of the glioma stem cells, by demonstrating the release of miR-1587 from glioma-associated mesenchymal stem cells [174]. This miRNA is able to inhibit the hormone receptor co-repressor-1 (NCOR1), a tumour suppressor of glioblastoma cancer stem cells, after extracellular vesicular transfers [82].

Indeed, chemoresistance to TMZ in glioblastoma could also be mediated by the presence of several CSCs, which are reported to be CD133+ cells. This population is associated with poor prognosis and resistance to chemotherapy [175]. In CD133+ cells deriving from T98G and U87MG tumours, miR-29a is significantly downregulated. The miR-29a overexpression improved sensitivity to cisplatin treatment in these cells and suppressed tumour growth in animal model of GBM [176]. A miRNA of the same family, miR-29c, has been demonstrated to be a sensitizer for TMZ treatment of resistant glioblastoma. Indeed, miR-29c targets specifically *Sp1*, thus reducing MGMT expression. In this way, the reduction of this enzyme allows the increased efficacy of TMZ treatment. The overexpression of miR-29c correlates with a good prognosis, suggesting the use of circulating miR-29c low levels as a prognostic biomarker of resistance [177].

The miR-423-5p, increased in glioblastoma, directly targets the nuclear protein Inhibitor of Growth 4 (ING-4). The inhibition of ING-4 leads to the upregulation of important signalling molecules, such as p-AKT and p-ERK1/2, strengthening glioma cell proliferation, angiogenesis, and invasion. The miR-423-5p overexpression, sustaining neurosphere formation and stemness, rendered glioma cells resistant to TMZ and increased p-AKT and p-ERK1/2 expression [178]. The same authors demonstrated that other miRNAs, let-7i, miR-151-3p, and miR-93, were downregulated in TMZ-resistant chinese GBM samples [179].

Recently, miR-126-3p expression has also been proposed as a modulator of GBM chemosensitivity. The authors found decreased expression of miR-126-3p in TMZ-resistant GBM tissues and cells. High levels of miR-126-3p, targeting *SOX2* mRNA and blocking Wnt/β-catenin signalling, enhanced TMZ sensitivity, inhibiting cell viability, reducing colony forming potential, and inducing apoptosis [180].

The same AKT/NF-κB pathway is also a final target of TMZ-upregulated miR-146b-5p [181]. By modulating the expression of tumour necrosis factor receptor-associated factor 6 (TRAF6) protein, miR-146b-5p enhanced the resistance against TMZ. TRAF6 is a signal transducer of TNFR family and Toll/interleukin-1 (IL-1) and its overexpression promotes the tumour cell proliferation in several solid tumours, i.e., breast, colon, and lung cancer [182]. TRAF6 is a known regulator of chemoresistance; it has been found upregulated in TMZ-resistant GBM tissues, and its overexpression increases the resistance of colon cancer cells to 5-Fluorouracil and to Bortezomib, respectively [183]. Overexpression of miR-146b-5p or TRAF6 knockdown significantly modulates the AKT/NF-κB pathway, decreasing the level of p-AKT and p-p65, activators of NF-κB [181]. The AKT/NF-κB pathway is involved in the regulation of autophagy, a process that emerging evidences suggests contributes to the development of TMZ resistance [184].

The pathway of PI3K/Akt is also target of miR-223: the exogenous expression of this miRNA promotes the survival of the cells exposed to TMZ by suppressing the PAX6 target gene, a known tumour suppressor in glioma cells [185,186].

In computational analyses it was shown that miR-93 and -193 could target Cyclin D1, a major regulator of cell cycle progression. Indeed, this protein regulates the progression of the cells from G1 to S phase. These two miRs also decreased cell cycling quiescence and induced resistance to TMZ [187].

The miR-7-5p has been described as a tumour suppressor in multiple cancers and it has been found significantly downregulated in TMZ resistant LN229 cells. Further experiments demonstrated that miR-7-5p has a role in the regulation of Yin Yang 1 (YY1), a transcription factor acting as a tumour promoter [188,189]. In glioblastoma, YY1 is able to interact with the NF-κB family member RelB, thus controlling both the proliferation of the cells as well as the pro-inflammatory cytokines leading to infiltration of glioma-associated macrophages [189]. The silencing of YY1 in cisplatin-resistant glioblastoma LN-229 cells was also obtained by overexpressing miR-186 which sensitizes the cells to drug treatment [190].

The miRNAs are also involved in the modulation of the hypoxia-induced autophagy process in different cancers. In glioblastoma, HIF1α increase and miR-224-3p decrease were simultaneously observed. It has been described that HIF1α knockout inhibited chemosensitivity by negatively regulating miR-224-3p expression under hypoxic condition, while hypoxia induced the expression of ATG5, a transcription factor possibly involved in the autophagic process. The researchers described a HIF-1α/miR-224-3p/ATG5 pathway that, in hypoxic condition, could influence the chemosensitivity of LN-229 cells to TMZ both in vitro and in the animal model [191].

Another miRNA involved in the control of autophagy is miR-519a. In U87 cells, made resistant to TMZ, it was found downregulated in comparison to the parental cell line. The exogenous overexpression of miR-519a dramatically enhanced TMZ-induced autophagy and apoptotic cell death through the inhibition of the STAT3/Bcl-2/Beclin-1 pathway [192].

The miR-138 is also upregulated in TMZ-resistant GBM, both in cell lines and in human tissue samples. BIM mRNA is the direct target of this miRNA, and miR-138/BIM axis promotes cell survival after TMZ treatment by regulating the autophagy process [193].

Some miRNAs are modulated indirectly by TMZ treatment: DNA methyltransferase (DNMT) is downregulated in TMZ-resistant GBM and this effect reduced miR-20a promoter methylation, leading to upregulation of miR-20a in TMZ-resistant cells. Vice versa, methyltransferase overexpression increases TMZ-sensitivity of GBM cells, reducing miR-20a expression [194].

Another miRNA involved in the control of both chemoresistance and metastasis is miR-139, being able to control glial fibrillary acidic protein (*GFAP*) gene expression. *GFAP* encodes for a protein of the cytoskeleton, mainly expressed in the astrocytes. This protein is highly expressed in high-grade brain tumours and involved in aggressive phenotype [195]. The polymorphisms of GFAP, especially those affecting the miR-139 tumour suppressor seed, have been found and correlated with the malignant phenotype. Indeed, the lack of miR-139 binding on *GFAP* mRNA increases the expression of GFAP protein, which in turn favours the assembly and stabilisation of the cytoskeleton and increases the chemoresistance to cytotoxic drug [196].

Possible changes in miRNA processing, such as the dysregulation of miR-221/222 processing by apurinic/apyrimidinic endonuclease 1 (*APE1*) mRNA, could also impact on chemoresistance process. APE1 is a member of the base excision repair process; it acts as a master regulator of the cellular response to genotoxic damage. In response to oxidative damage, APE1 stimulates the DNA repair by promoting the initiation of transcription of SIRT1, of the multidrug resistance gene (*MDR1*) and the phosphatase and tensin homolog (*PTEN*) tumour suppressor [197,198]. The miR-221 and miR-222, belonging to a polycistronic cluster, are involved in the regulation of PTEN expression, and have been already associated with chemoresistance in breast and in lung cancer [199,200,201]. APE1, being able to bind double-stranded RNA, can bind a precursor form of miR-221/222, and the silencing of APE1 results in decreased maturation of the miR-221/222 cluster. Upregulation of APE1, leading to an increase in miR-221/222 cluster expression, has been associated with the development of radio-resistance, cancer growth stimulation and invasion by inhibiting PTEN and with the development of multidrug resistance and altered response to chemotherapy [200,201,202,203]. Thus, it seems that interfering with APE1/miR-221/222 cluster regulation could impact on the chemosensitization of the GBM cells.

The oestrogen responsive miR-191 has been correlated with the aggressive mesenchymal phenotype of GBM [204]. Its expression was found to be altered by the aromatase inhibitor Letrozol in glioma cells [205]. Aromatase enzymes are necessary for the synthesis of the oestrogen hormone which, after binding to oestrogen receptor (ER) alpha and beta, interacts with oestrogen responsive elements (ERE) in the promoter region of miR-191, mediating the chemoresistance effect observed in several cancers and in GBM. Letrozole treatment reduces oestrogen-mediated binding of the ER to the ERE, reducing miR-191 expression. The evaluation of miR-191 function as a potential anti-tumoral drug and as a sensitizer of GBM cells should be still evaluated [206].

miR-101 is significantly downregulated in TMZ-resistant GBM cells (TMZ resistant forms of U251 and A172) and human samples and over-expression of miR-101 sensitizes resistant GBM cells to TMZ through the downregulation of glycogen synthase kinase 3β (*GSK3β*). This gene encodes for an enzyme involved in the regulation of protein synthesis, glycogen metabolism, cell proliferation and survival, and chemoresistance [207]. Several studies show that GSK3β plays a critical role in the development and progression of various malignancies and that targeting GSK3β may represent a novel strategy for the treatment of chemoresistant cancers [208,209].

An attempt of description of miRNA profiles able to select TMZ-sensitive from resistant GBM patients was proposed in two papers. In 2015, by in silico analysis, the authors proposed seven miRNAs, four of which (miR-1280, miR-1238, miR-938, and miR-423-5p) overexpressed in TMZ-resistant samples and three of which (let-7i, miR-151-3p, and miR-93) downregulated in TMZ-sensitive GBM human samples [179]. These miRNAs were used to classify an independent dataset of samples, demonstrating that two over seven are able to correctly classify TMZ-resistant patients. No hypothesis was made on possible miRNA targets, that could justify their different expression in TMZ-sensitive or -resistant cells. In 2016, another miRNA profile was proposed on U251 and A172, made resistant to TMZ. In this paper, the authors analyzed microarray results of these two cell lines and found common miRNAs, associated with TMZ-resistant phenotype. In particular, they found miR-9, -182 and -221 to be upregulated, while miR-29c, -93, -101, and -130a were to be downregulated. These results were confirmed in 12 TMZ-resistant human tumour samples compared to primary tumours. Focusing on miR-101, the in silico analysis of possible targets revealed that this miRNA could control the expression of GSK3β, VEGF, and COX-2. In particular, the authors demonstrated that miR-101 is able to downregulate the serine/threonine protein kinase GSK3β. The loss of miR-101, observed in TMZ-resistant cells, caused an increase in GSK3β expression, leading to drug resistance development and the inhibition of apoptosis. This regulation could be possibly due to the control of GSK3β on the methylation of the MGMT promoter. Loss of miR-101 was correlated with poor GBM patient prognosis [210].

## 4. In Vivo Imaging for the Study of GBM Heterogeneity and Microenvironment

Imaging techniques cover a pivotal role in the management of GBM patients [211]. In particular, in the clinical scenario the combinations of diffusion-weighted MRI (DWI-MRI) and amino acid-based PET techniques are challenging diagnostic procedures for the clinical management of patients. In addition, the multimodal MR-PET imaging offers new insights for a better understanding of GBM heterogeneity [212,213]. In a clinical setting, the conventional MRI sequences include: T1-weighted alone or with Gadolinium contrast agents (T1w-Gd) to evaluate the blood-brain barrier (BBB) permeability and tumour vascularization and T2-weighted and fluid attenuation inversion recovery (FLAIR) to define tumour oedema, border, and presence of necrosis [212]. Emerging techniques give complementary information. Diffusion-weighted MRI (DWI) provides information about the reorganization of cellular tissue and tumour invasion, by calculating the apparent diffusion coefficient (ADC) and its variant diffusion-tensor MRI (DTI) permits to distinguish the dominant white matter tracts through the tractographic information and to obtain and foresee the GBM invasive power. Perfusion-weighted imaging (PWI), with the perfusion acquisition modalities Dynamic susceptibility contrast (DSC) and Dynamic contrast-enhanced (DCE), furnishes data on cerebral blood flow and cerebral blood volume [214]. In addition, arterial spin labelling (ASL), using magnetically labelled arterial blood water protons as an endogenous tracer, is able to evaluate perfusion. Finally, MR spectroscopy (MRS) allows for the identification of the regional distribution of tissue metabolites [215]. However, despite advances in methodologies, the identification of necrosis, tumour margin, recurrence, and grading still represents a critical issue [216]. The use of PET and selected radiopharmaceuticals has allowed to image in vivo different biological features of tissue, such as glucose metabolism, cell proliferation, tissue hypoxia, inflammation, and matrix metalloproteinases that are associated with the tumour or tumour microenvironment. Here, are discussed the main cancer hallmarks relevant for GBM and TME and the potential use of radiopharmaceuticals for their in vivo measurement both at clinical and preclinical levels, after a research of the publications of the last five years in PubMed. Radiotracers and targets are summarized in Table 4. In the following paragraphs, different biological features of cancer and tumour microenvironment are described together with specific radiopharmaceuticals already available or under development for the in vivo imaging.

### 4.1. Cancer and Metabolism

Altered glucose metabolism is a characteristic of invasive cancers. Since 1927, Otto Warburg noticed that cancer cells preferentially converted glucose into lactic acid instead of pyruvate even in presence of sufficient oxygen, for this reason the “Warburg Effect” is also called aerobic glycolysis [272,273]. On this basis, 2-deoxy-2-[fluorine-18]fluoro-D-glucose ([^18^F]FDG) is the most used radiotracer in cancer; indeed, most cancers display increased glucose uptake and phosphorylation and there is a tight correlation between [^18^F]FDG uptake and cell viability [217]. Unfortunately, the brain tumour imaging represents a unique challenge because of the high background uptake due to normal grey matter which makes it very difficult to efficiently visualize with [^18^F]FDG PET [274].

Nevertheless, glioma shows several altered metabolic pathways which can be exploited with other markers to study cancer progression and to develop new metabolism-based therapeutic strategies [275]. Glutamine (Gln) is the second most abundant nutrient (after glucose) in blood circulation and supplies most of the carbon, nitrogen, and free energy and reduced equivalents necessary to support cell growth and division [276]. Gln is used by proliferating tumour cells to generate α-ketoglutarate which can be metabolised through the tricarboxylic acid (TCA) cycle to generate oxaloacetate (OAA) [277]. Tardito et al., analysing the metabolic role of Gln in sustaining the growth both in vitro (in human established GBM cell lines, in primary GBM stem-like cells, and in normal astrocytes) and in vivo (xenograft murine models and in GBM patients), found a tight collaboration between cancer cells and astrocytes. Indeed, Glutamine Synthetase (GS) is found in the majority of human GBM and its expression is associated with poor prognosis [278,279]. GS expression is greatly variable in tumour cells ranging from negative to high as in normal astrocytes. GS-positive astrocytes and/or GBM cells (potentially GSC) excrete Gln that supports the growth of GS-negative GBM cells [278]. Using imaging, Gln metabolism can be visualized with PET and ^18^F-(2S,4R)4-fluoroglutamine ([^18^F]F-Gln) or with Nuclear Magnetic Resonance (NMR) Spectroscopy [280]. In 2015, Venneti et al. characterised [^18^F]F-Gln uptake both in healthy mice and in several orthotopic murine models of glioma. Compared to [^18^F]FDG, [^18^F]F-Gln uptake was not observed in the brain of healthy mice and in animals with a permeable blood-brain barrier or neuroinflammation. On the contrary, [^18^F]F-Gln was able to identify very well both glioma margin in different murine models and tumor reduction after five days of treatment with TMZ. In addition, [^18^F]F-Gln avidity was observed in patients with the progressive disease [218]. The ability to visualize orthotopic glioma was also confirmed by other groups [219]. Clinical safety and pharmacokinetics in several tumours among glioma were also assessed. In particular, high-grade glioma with IDH mutation showed high uptake of [^18^F]F-Gln. Moreover, authors observed that [^18^F]F-Gln-avid tumours were uniformly [^18^F]FDG-avid but not vice versa [220].

At the moment, using NMRS is possible to quantify the abundance of the glutamate and glutamine pool defined as Glx and it has been observed an increase in Glx in high grade glioma or in invasive regions [281,282]. In other studies, authors have investigated the relationship between the presence of the IDH mutation and the presence of Glx and they found an inverse correlation between the presence of the oncometabolite 2-hydroxglutarate (2HG) and a significant decrease of Glx [283,284]. Heiland et al. associated the presence of specific metabolites to the subtypes of GBM on the basis of the gene mutations, in detail clustering of normalised Glx (nGlx) revealed two groups: one with high nGlx being attributed to the neural subtype and one with low nGlx associated with the classical subtype [285]. Considering the great importance to specifically identify Gln metabolism, Hangel et al. set up a reconstruction method, patch-based super-resolution (PBSR), in association with 7T MRSI to separate the peak of glutamate to that of Gln [286].

Additional metabolic pathways that have received considerable attention in GBM are fatty acid synthesis (FAS) and fatty acid ß-oxidation (FAO) [287,288]. In vivo studies performed in orthotopic mouse models of malignant glioma using ^13^C show that acetate contributes over half of oxidative activity within these tumours [289,290]. Kant et al. observed that FAO “high” phenotype was enriched in mesenchymal tumours, whereas proneural tumours were mainly characterized by FAO “low” phenotype indicating that a specific treatment could be used based on the GBM phenotype [287]. In addition, inhibition of either fatty acid synthesis or beta-oxidation reduces proliferation of glioma cells [291,292]. [^11^C]Acetate is a radiotracer that is related to enhanced lipid synthesis and it is rapidly transported into cells through monocarboxylic acid transporters (MCTs) and metabolised into acetyl-CoA [293]. Kim et al. evaluated that [^11^C]Acetate PET/CT is able to differentiate between low- and high-grade glioma and, specifically, grade III from IV. Authors also found a correlation between the expression of acetylcoenzyme A synthetase (ACSS2), glioma grade and prognosis indicating ACSS2 as a potential drug target [221]. Finally, in a preclinical study Koyasu S et al. reported that U251 tumours carrying a mutation in IDH1 gene (R132H) display a higher uptake of [^14^C]Acetate compared to U251 wt tumours [222].

In recent years, the tryptophan (TRP) metabolic pathway emerged as an important node both for the GBM progression and for resistance to therapy because it actively contributes towards immune suppression in particular in mesenchymal and classical molecular subtypes [294]. Tryptophan metabolism appears to play the most dominant role in immune tolerance in GBM, with over a two-fold increase in Tregs and a strong trend in decreased CD8+ T cells. TRP is an essential amino acid necessary for protein synthesis. The bulk of TRP is therefore available for metabolism via four pathways to produce physiologically important metabolites including serotonin, melatonin, tryptamine, and kynurenine (KYN). Over 95% of TRP is metabolized to KYN via the kynurenine pathway (KP). Through local TRP depletion and the production of the toxic metabolites, the KP fosters an immunosuppressive tumour microenvironment [295,296]. Three distinct enzymes are able to complete the initial and rate-limiting step of the metabolism of TRP to N-formylkynurenine (NFK): indoleamine 2,3-dioxygenase 1 (IDO1), indoleamine 2,3-dioxygenase 2 (IDO2), and tryptophan 2,3-dioxygenase 2 (TDO2) [224]. IDO1 is expressed in a wide variety of cell types, including dendritic cells, endothelial cells, macrophages, fibroblasts, and mesenchymal stromal cells, as well as in neurons and in cancer cells themselves [297]. Moreover, as observed by Ozawa et al., IDO1 is also highly expressed in glioma stem cells [298]. Given the importance of the kynurenine pathway in numerous cancers, it is increasingly recognised as a prominent target for cancer imaging and therapy. Currently, radiotracer alpha-[^11^C]methyl-L-tryptophan ([^11^C]AMT) is used for PET imaging of the IDO-mediated kynurenine pathway in clinical and preclinical research [223,224]. In two studies, Bosnyak et al. evaluated the ability of [^11^C]AMT to differentiate between meningioma and grade II and grade III gliomas and the prognostic role of the radiotracer, finding that GBM patients with higher pre-treatment tumoral tryptophan uptake, expressed as tumour/cortex SUV-ratios, showed longer survival [225,299].

In two different studies, John et l. have used combined MRI and [^11^C]AMT PET for discriminating non-enhancing GBM regions which are challenging to detect with MRI only. They have suggested the importance of the amino acid uptake as a key player in distinguishing metabolically active subregions in the tumour from the areas with necrosis and oedema, which correspond, respectively, to enhancing and non-enhancing tumour zones. In fact, GBM showed very heterogeneous regions within the tumour with high-uptake regions often extending into the non-enhancing brain with high cellularity and the non-resection of these areas predicted the site of post-treatment progression. Furthermore, they have suggested the high uptake of tryptophan in contrast-enhancing subregions as an independent imaging biomarker for detecting the prolonged survival in newly diagnosed GBM patients [212,226]. Guastella et al. observed that PDXs generated by injection in mice of GBM cells or tumour fragments derived from patients with different uptake of [^11^C]AMT and expression of TRP pathway markers recapitulate the same TRP metabolic characteristics of the primary patient tumour underling as these models can be used to test new therapeutic approaches [224].

Nevertheless these interesting results, the use of [^11^C]AMT is limited to a few PET centres due to its short half-life (20 min) and the complex radiosynthesis [300]. So, a number of fluorine-18 labelled tryptophan analogues have been developed for cancer imaging, among them L-1-[^18^F]fluoroethyl-tryptophan (L-1-[^18^F]FETrp). L-1-[^18^F]FETrp uptake was tested in several cancer cell lines, both in vitro and in vivo, including 73C-glioma cells. Intracranial brain 73C-glioma transplanted mice confirmed the capability of L-1-[^18^F]FETrp to pass the BBB and to efficiently visualize the tumour suggesting that L-1-[^18^F]FETrp is a promising radiotracer for PET imaging of cancer [227].

The most widely used radiolabelled amino acid is the essential amino acid methionine, labelled with ^11^C, which is taken up by the L-type amino acid transporters (LAT) 1 (SLC7A5) and 2 (SLC7A8) and incorporated into the protein. [^18^F]fluoroethyltyrosine (FET) is a substrate for LAT1 and LAT2 and is widely used in place of methionine considering that it has a longer half-life (109 min) compared to that of C-11 (20 min) [301]. Both radiotracers are able to visualise low and high glioma but with difficulty it is possible to differentiate them. The radioligand [^18^F]3′-deoxy-3′-fluorothymidine (FLT), substrate of thymidine kinase that is overexpressed during S-phase of cell cycle, reflects cell proliferation [302]. Since [^18^F]FLT does not cross intact the BBB, it does not accumulate in low-grade tumours or stable lesions but it is detectable in high-grade (grade III or IV) tumours with a disrupted BBB [303]. It is used for imaging early treatment response and for predicting clinical outcome in brain tumours [231]. It is widely used in preclinical research to evaluate early response to standard or new therapeutic approaches [232,233,234,235,236].

Because of enhanced cell proliferation and resulting elevated levels of cell membrane synthesis during tumorigenesis, choline metabolism is also involved in brain cancer. Choline is a precursor for the synthesis of the phospholipid components of the cell membrane. Cellular choline is phosphorylated by choline kinase (CK) yielding phosphocholine (PCho), which further reacts with CTP to yield CDP-choline. The de novo synthesis (Kennedy pathway) of phosphatidylcholine (PC) then results from the reaction of CDP-choline with diacylglycerol. In the clinical setting, two imaging modalities allow for the examination of choline met abolism, namely proton magnetic resonance spectroscopy (^1^H-MRS) and PET using [^11^C]Choline. MRS provides non-invasive measurements of tissue concentrations of metabolites, such as total choline-containing compounds (tCho), total creatinine (tCr), and N-acetylaspartate (NAA). Increased tCho levels result from elevated cell membrane turnover and cellular density [237].

Wehrl et al. performed a comparative in vivo study assessing tCho levels by MRS using chemical shift imaging (CSI) and [^11^C]Choline metabolism by PET with morphologic parameters, in a murine orthotopic glioma mouse model obtained with murine SMA-560 (spontaneous murine astrocytoma) glioma cells. Authors found an increase in [^11^C]Choline uptake in the tumour paralleled to a decrease in tCho levels. Only a small overlap between tumour volumes, as identified by CSI and PET, was found; in detail, CSI highlights the areas of high choline concentration, which seems to be localized to the tumour rim, whereas [^11^C]Choline PET identifies the regions of high choline turnover. Therefore, MRS and PET seems to give complementary information about choline metabolism [237].

MRS and DTI-MRI are used to identify metabolic changes in the invasive margin of glioblastomas and to monitor therapy response [281,304]. In a recent study, Takei et al. evaluated if PET with multi tracers was able to differentiate glioma according to the 2016 World Health Organization (WHO) classification of tumours of the CNS. [^11^C]Methionine, [^11^C]Choline and [^18^F]FDG uptake was higher in the anaplastic astrocytoma (AA) IDH-wt group than in the IDH-mut group, whereas in GBM only [^11^C]Methionine and [^11^C]Choline uptake were significantly higher in IDH-wt group than in IDH-mut group suggesting that PET is able to differentiate IDH-wt and IDH-mut tumours [238].

### 4.2. Cancer and Hypoxia

Hypoxia is a key driver of tumour adaptation promoting tumour progression and resistance to therapy. In particular, hypoxia and hypoxia-related factors, such as HIF-1α, are associated with pseudo-palisading necrotic regions that protect stem cells niche, contributing to the aggressive profile of these tumours. HIF-1α protein expression in glioblastoma promotes angiogenesis by inducing VEGF [305]. For these reasons, it is pivotal to have therapies that counteract this situation and imaging techniques that allow one to identify hypoxic regions within the tumour. In a systematic review, Raccagni et al. perform an overview on PET radiotracers dedicated to visualise hypoxia including [^18^F]-labelled nitroimidazole compounds ([^18^F]FMISO and [^18^F]FAZA), radiolabelled copper-ATSM complexes, Carbonic anhydrase 9 (CAIX) agents, and new generation tracers such as [^18^F]EF5 and [^18^F]EF3, [^18^F]3-NTR and [^18^F]HX4 [239]. Among these, [^64^Cu]ATSM results are very interesting for GBM. This radiotracer can be used both as an imaging and as a therapeutic agent, and it is considered as an independent prognostic biomarker of survival in high-grade glioma [240,241]. ^64^Cu is a promising theranostic radionuclide owing to its suitable half-life (12.7 h) with multiple decay modes, including β + (18%), which is used for PET imaging, β − (39%, 0.95–1.4 mm tissue range), and Auger electron (43%, ~126 nm tissue range), which are used for therapeutic radiation. Jin et al. in a GBM mouse model obtained with the injection in the right flank of U87MG human glioblastoma cells evaluated the intra-tumour distribution of [^64^Cu]-cyclam-RAFT-c(-RGDfK-)_4_ ([^64^Cu]RaftRGD), a radiotracer for α_V_β_3_ integrin and [^64^Cu]ATSM and their therapeutic efficacy. Mice were treated with 18.5 or 37 MBq of [^64^Cu]RaftRGD or [^64^Cu]ATSM, or a combination of both at a dose of 18.5 MBq for each agent. Interestingly, the distribution of the two radiotracers was different but complementary. The single administration of 18.5 MBq did not show any significant effect on tumour growth, whereas the combination of both at a dose of 18.5 MBq inhibited tumour cell proliferation and prolonged mice survival versus the single treatment with 37 MBq [242]. Similarly, Yoshii et al. in the same mouse model, tested the efficacy of different single doses of [^64^Cu]ATSM (18.5, 37, 74, 111, or 148 MBq or saline as a control) and of multiple administrations (37 MBq × 4). Multiple administrations significantly inhibited tumour proliferation and increased survival also compared to the highest single doses, without causing haematological toxicity [243]. Recently, Peres et al. examined the biodistribution of [^18^F]FMISO and [^64^Cu]ATSM in an orthotopic GBM model obtained with the inoculation of C6 cells in male Wistar rats. [^64^Cu]ATSM displayed a higher uptake in the tumour compared to [^18^F]FMISO and also a higher extension of the signal. [^64^Cu]ATSM localised in hypoxic regions where there was expression of pimonidazole marker, but also in non-hypoxic regions with high expression of CD68 and GFAP used to assess inflammation and astrogliosis [244].

### 4.3. Cancer and Inflammation

A large proportion of the tumour microenvironment consists of inflammatory infiltrate predominated by microglia and macrophages, which are thought to be subverted by glioblastoma cells for tumour growth [306]. Thus, GAMMs are logical therapeutic targets [307]. In addition to GAMMs, other inflammatory cells are present, such as tumour-infiltrating lymphocytes, neutrophils, and cancer-associated fibroblasts [308]. Inflammatory cells promote tumour growth, invasion, and tumour-to-therapy resistance thanks to the release of specific molecules and factors that favour anti-inflammatory activity (TGF-ß, ARG1 and IL10), tissue remodelling and angiogenesis (VEGF, MMP2, MMP9). In addition, inflammatory cells influence glioma-stem cells through the release of the matrix metalloproteinase MMP9 [309].

The translocator protein (TSPO; 18 kDa) is a peripheral benzodiazepine receptor, composed of a transmembrane multimeric protein complex of 18-kDa, situated in the outer mitochondrial membrane. TSPO is widely distributed in most peripheral organs. In addition, TSPO is also minimally expressed in resting microglial cells in the healthy brain but is substantially upregulated in reactive astrocytes and predominantly during the microglia activation process, due to neurodegenerative and neuroinflammatory diseases [310]. Moreover, most glioma cells express the TSPO. Different studies demonstrated a positive correlation between TSPO expression and grade of malignancy and a negative correlation with survival [311]. The most used TSPO radiotracer is [^11^C](R)PK11195. Su et al. found that [^11^C](R)PK11195 uptake in high-grade gliomas was significantly higher than in low-grade astrocytomas and low-grade oligodendrogliomas. TSPO in gliomas was expressed predominantly by neoplastic cells, and its expression correlated positively with uptake in the tumours indicating that [^11^C](R)PK11195 can be used to stratify patients [245]. Considering the short half-life of ^11^C-labelled compounds, other new generations TSPO specific ligands have been developed, such as [^18^F]DPA-714 and [^18^F]GE-180. In a human glioma-injected mouse model, obtained after the injection in the brain of Gli36dEGFR-LITG, Zinnhardt et al. performed multi tracer imaging PET studies ([^18^F]FET, [^18^F]DPA-714, and [^18^F]BR-351, this last for MMPs) to evaluate the inflammatory tumour microenvironment. Authors found that the three radiotracers only partially overlapped; glioma cells overexpressed TSPO contributing to the most of [^18^F]DPA-714 uptake, but also infiltrating glioma associated myeloid cells (GAMs) showed an important TSPO and MMP expression [246]. In another study performed in patients with glioma, authors found a strong relationship (r = 0.84, *p* = 0.009) between the [^18^F]DPA-714 uptake and the number and activation level of GAMs. TSPO expression was mainly restricted to human leukocyte antigen D related-positive (HLA-DR+) activated GAMs, particularly to tumour-infiltrating HLA-DR+ myeloid-derived suppressor cells and tumour-associated macrophages indicating that TSPO radiotracer could be useful to study tumour microenvironment [247]. At the same time, in a preclinical model obtained with human GBM cells from patient, [^18^F]DPA-714 was able to detect glioma infiltration into the contralateral brain earlier than [^18^F]FET. In parallel, DWI MRI provides microstructure information. Authors also found an abundance of CD11b+ GAMs inside and around the tumour [248]. We have recently developed and characterised a new TSPO radiotracer, [^18^F]VC701 in neuroinflammation models [249]. Similarly, in an orthotopic glioma model obtained with murine GBM cells, the GL261 ones, we observed an earlier TSPO uptake signal compared to [^18^F]FLT uptake (Figure 2).

Furthermore, the radiotracer [^18^F]GE-180 seems to provide interesting results in patients and the uptake correlates with the grade of tumour [250,251]. Other radiotracers are now under development to increase the specificity to TSPO [252,253].

### 4.4. Other Targets of Tumour Microenvironment

Prostate-specific membrane antigen (PSMA) is a type II transmembrane glycoprotein receptor with glutamate carboxypeptidase/folate hydrolase activity [312]. PSMA is highly expressed in prostate cancer and has recently emerged as a target for radionuclide imaging and treatment of this tumour [313]. PSMA is expressed not only in prostate cancer, but also in other solid tumours including brain tumours and correlates with the WHO grade. PSMA expression was found in endothelial cells associated with neovascularization and, for these reasons, can be a very interesting marker for monitoring anti-angiogenic drugs [314]. Several PET radiotracers for PSMA have been developed, such as [^68^Ga]PSMA-HBED-CC, [^18^F]DCFPyL, and [^89^Zr]Df-IAB2M anti-PSMA minibody. At the moment, only few studies evaluating PSMA targeted imaging of gliomas/glioblastomas in clinical practice are currently available. Moreover, most of these studies are case reports or case series [255,256,257,258,259,260,261,262,263,264,315]. Although limited, these studies confirmed that high-grade glioma/glioblastoma are PSMA-avid compared to low-grade and PSMA radioligand mainly binds vascular endothelial cells. A more detailed review is present in the paper of Van de Wiele et al. [265]. This year, a preclinical study in which the authors compared the uptake of two PSMA radioligands ([^68^Ga]-PSMA and [^18^F]DCFPyL) in three rat glioma models (F98, 9L, or U87) has been published. Radiotracers’ uptake was evaluated ex vivo with autoradiography and in vivo with PET. Uptake data were then correlated to immunofluorescence staining for blood vessels, microglia, astrocytes, and the presence of PSMA. Both radiotracers displayed a higher uptake at the rim of the tumours where CD11b staining demonstrated low presence of activated microglia but a strong GFAP staining indicating the presence of reactive astrocytes. Very few astrocytes were noted in the centre of the tumours. These data suggest that PSMA radioligand could also visualize astrocytes activation [266].

Matrix metallo-proteinases (MMPs) are other attractive biomarkers for tumour therapy and imaging. MMPs are linked to increased cell proliferation, tumour invasion, migration, and poor prognosis in glioma patients [316]. Moreover, MMPs facilitate microglia-mediated glioma invasion by degrading the basement membrane and proteins of the ECM. Besides that, MMPs affect the neuroinflammatory milieu by modulating the expression and activity of chemokines, inflammatory cytokines, growth factors, and receptor turnover [317]. Several molecules have been recently developed to study MMPs. For example, de Lucas et al. evaluated an immunoPET tracer for the membrane-type 1 matrix metallo-proteinase (MT1-MMP or MMP-14), [^89^Zr]DFO-LEM2/15, in xenograft and orthotopic brain GBM models. This tracer displayed the higher uptake after 24 h post injection in tumours with high expression of MT1-MMP but a severe disruption of the BBB is needed to visualise intra brain tumours [267]. Zinnhardt and colleagues, in a multiple tracers study to evaluate the inflammatory microenvironment in an orthotopic glioma model, used the MMP inhibitor compound (R)-2-(N-benzyl-4-(2-[^18^F]fluoroethoxy)phenylsulphonamido)-N-hydroxy-3-methylbutanamide ([^18^F]BR-351), which binds to the activated forms of MMP-2, -8, -9, and -13. Areas of exclusive [^18^F]BR-351 were observed at the outer borders of the tumour volume and the area of radiotracer uptake agreed with in situ zymography, indicating elevated levels of activated MMP-2 and MMP-9 [246]. Wang et al. synthesized a membrane-penetrating cyclic peptide, named iCREKA, labelled by fluorescein isothiocyanate (FITC) and positron emitter ^18^F. The cyclic peptide iCREKA reaches tumour tissues via blood circulation. CREKA is expected to specifically bind to the fibrin-fibronectin complexes that are widely and abundantly distributed in tumour stroma. MMP-2/9 recognize and cleave iCREKA between CREKA and the membrane-penetrating peptide, while hydrolytic enzymes hydrolyze the disulfide bond. As a result, the fluorescent or radionuclide-labelled membrane penetrating peptide is released, which enters the plasma membrane and the tumour cells. The microPET/CT imaging demonstrated that [^18^F]iCREKA could target U87MG xenograft tumour in vivo from 30 min to 2 h after injection [268]. Zhao et al. investigated the biodistribution and uptake of [^18^F]-fluoropropionyl-chlorotoxin ([^18^F]-FP-chlorotoxin), which binds to MMP-2 in an orthotopic rat C6 glioma model. The uptake of the tracer in the normal brain is very low and a high accumulation was found in the glioma tissue. The tumour to normal brain ratio of [^18^F]-FP-chlorotoxin was higher than that of [^18^F]FDG with the maximum uptake at 90 min [269]. Kasten and colleagues designed an MMP-14-activatable dual PET/NIRF peptide probe for imaging and guiding resection of glioma. The peptide probe combined (1) a NIRF reporter and quencher pair separated by a peptide sequence (MMP-14 “substrate peptide”) that is cleaved specifically by activated MMP-14 to release the quencher and allows visualization of the NIRF dye; and (2) a chelate for radionuclides attached to a peptide sequence, that binds to MMP-14 (MMP-14 “binding peptide”) and enables PET imaging. For PET imaging the peptide was labelled with [^68^Ga] or [^64^Cu] and biodistribution was evaluated in an orthotopic glioma model obtained with PDX JX12 tumour. Both [^68^Ga]binding-peptide and [^64^Cu]binding-peptide allow to visualize PDX tumours. PET and NIRF signals correlated linearly in the orthotopic PDX tumours and the signals co-localized with MMP-14 expression identified with immunohistochemistry [270].

A further PET radiotracer can be the Fibroblast Activation Protein (FAP) ligand.

Recently, Röhrich and colleagues have shown the potential of the targeting FAP-PET-based imaging (^68^Ga-FAPI-01 and -04) in retrospectively studying 13 patients with wild type-IDH glioblastomas. FAP-positive signalling of extracranial tumours is attributed to activated Cancer Associated Fibroblasts (CAFs) that are located in the stromal compound of these tumours and overexpress FAP. In parallel, patients underwent multimodal MRI. FAP specific PET and rCBV MRI scans are modestly correlated and no correlation between the FAP-based PET and the ADC sequence has been observed, suggesting that FAP-specific PET could be useful in biopsy organization and discrimination between pseudo-progression and tumour progression after radiation therapy for GBM imaging [318].

In a paper of Henderson and colleagues, two recurrent GBM patients have been chosen for validating [^18^F]Fluciclovine-based PET/CT imaging in GBM, a synthetic amino acid used as PET tracer. In this study, it has been reported the ability of this radiotracer, approved by FDA for prostate cancer care in 2016 and used since 2015 for gliomas treatment but not still approved, to be feasibly used for discriminating areas with the highest cancer recurrence from the ones where the most prevailing treatment efficacy is observable [228].

Considering the importance of the L-amino acid transporter 1 (LAT1), other radiotracers have been exploited and investigated.

In a paper by Verhoeven et al., the radiotracer 2-[^18^F]-2-fluoroethyl-l-phenylalanine (2-[^18^F]FELP), has been synthesized, tested both in vitro and in vivo in F98 models and compared to [^18^F]FDG and [^18^F]FET. In vitro, 2-[^18^F]FELP has shown the highest affinity and a lower uptake in presence of one of the LAT1 inhibitors compared to the [^18^F]FET uptake, suggesting a specific LAT-1-mediated 2-[^18^F]FELP transport. Moreover, a similar trend in the in vivo data was observed. To recapitulate, being LAT1 overexpression a GBM distinctive marker and having 2-[^18^F]FELP a very high and specific affinity for this amino acid transporter, this emerging tracer might be exploited for driving biopsy and primary brain tumour diagnosis, to plan and guide radiation treatment and for tumour recurrence and radio-necrosis discrimination after the starting therapy [229].

Another LAT1-specific radiotracer, the (S)-2-amino-3-[3-(2-^18^F-fluoroethoxy)-4-iodophenyl]-2-methylpropanoic acid (^18^F-FIMP) has been synthetized and validated in LAT1-positive human glioblastoma cells and tumours. [^18^F]FIMP radiotracer specifically accumulated in LAT1-positive tumour tissues and not in inflamed tissue. Moreover, ex vivo analysis confirmed the stability of this radiotracer [230].

### 4.5. miRNA and Imaging

Several years of research correlating imaging features with histopathological data created a new branch of medicine: radiomics. Radiomics is an analysis approach to convert the image data into quantitative, mathematical output (features). Radiomics features are closely related to intra-tumour complexity and heterogeneity, representing the histopathological characteristics of the GBM. Imaging techniques could be helpful for tumour classification, analysis of the response to the treatment, and prognosis [319]. Radiogenomics is an emerging field that tries to link radiomics with the corresponding underlying genomic and epigenomic data. Radiogenomics analysis highlights the association between the radiomic quantitative data/features and molecular profile, including genomic, epigenomic, and metabolomic signatures, that are correlated with a clinical outcome [320]. The radiogenomics approach has been proposed for some types of cancer, from hepatocellular carcinoma to breast cancer [321,322]. Genomic data acquisition requires tissue biopsy, which is not always obtained in patients with GBM. Histopathological features, specific mutation pattern and molecular markers analysis revealed that GBM could be classified in four subtypes with different outcomes: proneural (the most favourable for the prognosis) neural, mesenchymal (the one associated with poor prognosis), and classical. Different subtypes react differently depending on treatment [323]. Radiogenomics can be applied to predict the GBM subtypes. Volumes of both contrast enhancement and necrosis are higher in GBM of the mesenchymal subtype compared to the proneural subtype. GBMs with less than 5% tumour enhancement are mostly of the proneural subtype.

Radiogenomics could also help in the prediction of the mutation status. Indeed, the IDH-mutant GBM is mainly localised in the frontal lobe and it is characterized by a higher percentage of non-contrast-enhancing part of the tumour and the presence of cysts on MRI. Left temporal lobe GBM are usually MGMT-methylated tumours, whereas MGMT-unmethylated GBM are more frequently found in the right hemisphere. 1p19q co-deletion is linked to classical oligodendroglial MRI characteristics (heterogeneous T2 signal intensity and calcifications). Tumours with EGFR amplification, mainly localised in the left temporal lobe, present a significant higher percentage of contrast-enhancement and T2/FLAIR hyperintensity compared with those lacking EGFR amplification. The relation between MRI features and the expression of other molecular markers (phosphatase and tensin homolog, PTEN; platelet-derived growth factor receptor, PDGFRA; cyclin-dependent kinase inhibitor 2A, CDKN2A; retinoblastoma 1, RB1; tumour protein 53, TP53) is under investigation [213].

Some attempts have been made in the description of GBM heterogeneity on the basis of epigenetic profile (DNA methylation), associating it with patients’ survival, MRI, and quantitative pathology (tumour morphology, proliferative activity, and microenvironment). From this work it emerges that mesenchymal cell subtype, associated with the worst prognosis, is enriched in hypomethylated promoter regions in EZH2, KDM4A, RBBP5, and SUZ12, as well as in regulators of pluripotency (NANOG, SOX2, POU5F1) and enriched in hypermethylated chromatin promoter of CTCF, EZH2, and KDM4A. MRI data reveals that the highest number of immune cells was in tumours of the mesenchymal subtype, showing also increased tumour size, fewer vital tumour areas, and increased oedema, while histopathology revealed lower cell density and large necrotic areas [324].

In 2012, Elkhaled et al. demonstrated that there is concordance between the presence of D-2-hydroxyglutarate (2HG), as detected by proton high-resolution magic angle spinning (^1^H HR-MAS) spectroscopy, and *IDH1* mutation status, as determined by antibody staining and genetic sequencing, that was confirmed by Choi and colleagues [325,326]. These two studies revealed that there is a direct link between a genetic mutation and a behavioural or a metabolic phenotype observed by imaging analysis.

In 2018, a paper described a tentative correlation of data from quantitative MRI with transcriptomic profiles of 65 patients with primary glioblastoma [327]. Unsupervised clustering methods, confirming the data of Verhaak et al. revealed the existence of three different clusters, each associated with a different outcome [327,328]. Each cluster represented a distinct molecular classification of glioblastoma already described, classical type, proneural and neural types, and mesenchymal type. The first cluster was associated to the increased expression of anion channel activity, peroxisome, and the classical phenotype observed in Verhaak and colleagues’ paper; the second cluster presented higher expression of genes of the mitosis, cell cycle and a proneural/neural subtype; the third cluster was characterized by altered expression of extracellular matrix molecules, defence response, immune signalling molecules, and the mesenchymal subtype molecules. These data suggest that radiomic data could provide information about molecular subtypes of GBM based on the transcriptomic profile [327,328].

Nowadays, a missing step is to demonstrate the existence of a correlation between the data extracted from MR imaging and the miRNA expression profiles.

Molecular imaging by the use of PET is increasingly being implemented into clinical practice for treatment planning and response monitoring in GBM. The potential clinical applications of PET include the monitoring of treatment response, as it could distinguish tumour recurrence from radiation necrosis or pseudoprogression. Correlation between different types of amino acid tracers in PET and molecular markers is currently under investigation [329].

A correlation between miRNA expression in tissue GBM and PET imagings has not been proven yet. Indeed, miRNAs free molecules or miRNAs associated to extracellular vesicles or exosomes released by GBM tumour have been considered as possible biomarkers in the liquid biopsy for GBM diagnosis [213].

Several miRNA profiles have been proposed for GBM diagnosis by liquid biopsy isolation (i.e., as proposed by Kopkova and colleagues and by Ebrahimkhani et al.), but no one has correlated their expression with PET imaging features [330,331].

Exosomal miR-21 is one of the plasma biomarkers proposed for GBM diagnosis and seems to be able to distinguish tumour progression from pseudoprogression or radionecrosis [332].

Recently, a noninvasive glioblastoma testing (NIGT) platform has been proposed, including a multimodal approach that combines the complexity and heterogeneity of GBM described by imaging techniques, with computational approaches, and the use of circulating biomarkers [213]. This could be especially helpful in the selection of patients in order to better address them to the best therapeutic approach.

## 5. Final Considerations and Future Direction

In this review, starting from the analysis of GBM cell heterogeneity, we focused on those aspects that sustain the development of TMZ resistance, in order to find out possible therapeutic solutions for clinical application. In particular, we highlighted the supporting role of microenvironment surrounding the tumour and the possible functions of several non-coding miRNAs, as possible targets for the development of new therapeutic strategies. In order to have a complete view of the gold standard treatment of GBM, we described the canonical treatment of TMZ and the main mechanisms involved in the development of resistance to TMZ. In addition, several possible alternative treatments to TMZ alone-based therapy have been discussed, including combinatorial pharmacological treatments, comprising radiotherapeutic approaches, antibodies, and miRNAs. To the view of the interesting role of miRNAs, new therapeutic strategies could include the combination of miRNAs and drugs, this field becoming an important opportunity to investigate [109,333,334,335,336].

Moreover, we discussed a significant number of PET tracers that can help to visualise and monitor cancer cells and tumour microenvironments including GAMMs, endothelial cells, fibroblasts, astrocytes, and stem cells (Figure 3). These PET radiopharmaceuticals, particularly when associated with information deriving from MRI, may be relevant also for the identification of lesion heterogeneity and for the clusterisation of subsets of patients potentially responding to treatment under development. In vivo diagnostic imaging data may be a useful platform to associate and monitor genetic and biological features of the tumour at sub-regional levels during time in a non-clinical model or associated with post-surgery analysis and outcome in patients. In turn, in vivo molecular imaging offers a potential tool to characterize the inter- and intra-tumour heterogeneity of GBM and clusterize the lesions. Moreover, new advanced technologies and analysis methods as combied PET/MRI scanners joined by textural features increase diagnostic accuracy of gliomas in the identification of specific genotypes guiding surgery.

In addition, PET tracers can be used for guiding targeted radionuclide therapy substituting the positron emitting radionuclide with a therapeutic beta-particle emitting radionuclide, either lutetium-177 or yttrium-90 as performed in other types of tumor. Although this application is not yet available or proof for glioma, the use of theranostic [177Lu]-PSMA has been already tested in brain metastasis indicating the potential feasibility for its application also in glioma [337].

## 6. Conclusions

In this review we have summarised the complex heterogeneity of GBM and the potential use of imaging to visualise differences at regional levels in biological features. We have described the main genetic and epigenetic mechanisms of drug resistance and the state-of-the-art research studies attempting to overcome it. It should be stressed that surgery remains the best treatment available. However, the delayed diagnosis and difficult identification of lesion margins and infiltrating cells limit its benefit. For these reasons, pharmacological as well as radiation therapies are necessary. Despite the efforts, information on the biology of GBM is still insufficient and the molecular classification limited to a selected portion of tumour. However, molecular and single-cell data suggest that treatment should act on common features, such as the metabolism or immune system, particularly using combined strategies. In this scenario, molecular imaging may help to better define GBM complexity and the modification occurring during therapy at regional level. This, together with biopsy or post-surgery in vitro single-cell imaging and bioinformatics, can help in the identification of cluster of patients with common biological traits and this is crucial for the proper development of novel pharmacological or radiation-based therapy.

## Figures and Tables

**Figure 1 ijms-21-05631-f001:**
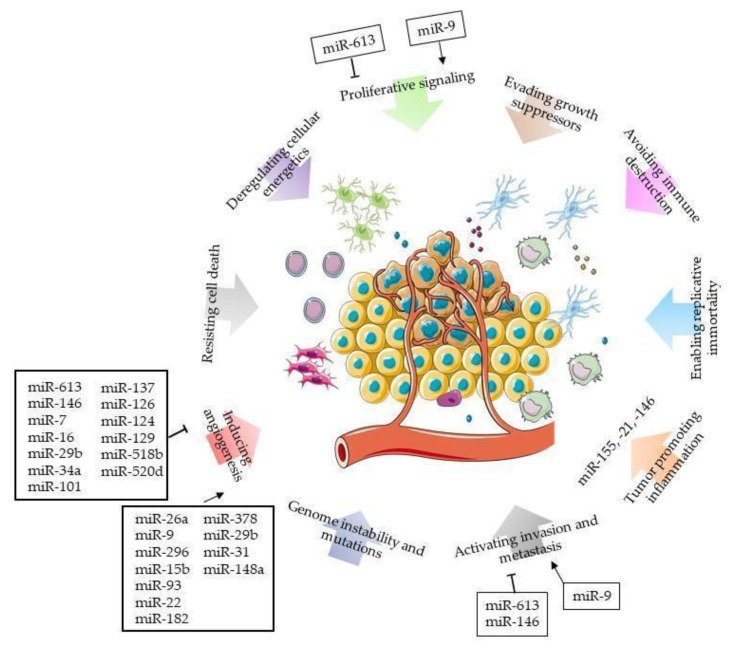
Role of miRNAs in GBM.

**Figure 2 ijms-21-05631-f002:**
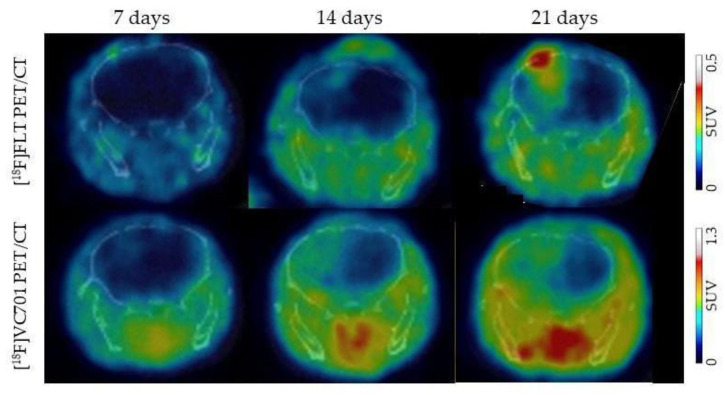
Representative [^18^F]FLT and [^18^F]VC701 PET/CT images of a mouse bearing GL261 glioma tumour after 7, 14, and 21 days from cell injection. [^18^F]VC701 uptake is visible before [^18^F]FLT signal. Colour scale is expressed as Standardized Uptake Value (SUV).

**Figure 3 ijms-21-05631-f003:**
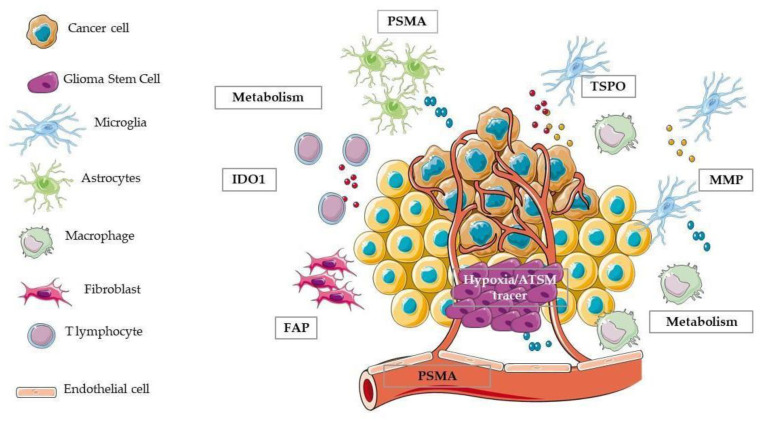
Targets of radiopharmaceutical used for in vivo imaging of tumour microenvironment in glioma.

**Table 1 ijms-21-05631-t001:** Some of the miRNAs involved in angiogenesis in Glioblastoma (GBM) (publications of the last 5 years).

miRNA	GBM Expression Level	Function	Target	References
miR-7	Down	Inhibits angiogenesis	*EGFR*, *IRS-1*, *IRS-2*, *FAK*, *OGR*	[76]
miR-296	Up	Promotes angiogenesis	*HGS*	[99]
miR-15b	Up	Inhibits angiogenesis	*NRP-2*	[100,101]
miR-93	Up	Promotes angiogenesis	Integrin B8	[102]
miR-16	Down	Inhibits angiogenesis	*BMI-1*	[103]
miR-613	Down	Inhibits angiogenesis	*VEGFA*	[84]
miR-26a	Up	Promotes angiogenesis	*PTEN* and *PI3K/Akt*	[85]
miR-9	Up	Promotes angiogenesis	*COL18A1*, *THBS2*, *PTCH1* and *PHD3*; *HIF-1α*/*VEGF*	[86]
miR-9-5p, miR-22-3p, miR-182-5p	Up	Promotes angiogenesis	*RGS5*, *SOX7*, and *ABCB1*	[104]
miR-29b, miR-34a, miR-101, and miR-137	Down	Inhibits angiogenesis	stanniocalcin-1 (STC1) induces *eNOS*, *VEGF*, and *VEGFR2*	[105]
miR-126/126-3p	Down	Inhibits angiogenesis	epidermal growth factor-like protein 7 (EGFL7)	[106]
miR-124-3p	Down	Inhibits angiogenesis	NRP-1/GIPC1 pathway	[107]
miR-129-5p	Down	Inhibits angiogenesis	*WNT5a*	[108]
miR-378	Up	Promotes angiogenesis	*VEGFR2*	[109]
miR-518b	Down	Inhibits angiogenesis	platelet-derived growth factor receptor β (*PDGFRB*)	[110]
miR-29b	Up	Promotes angiogenesis	*BCL2L2*, which in turn regulates *Ang-2* and *VEGF*	[111]
miR-520d-5p	Down	Inhibits angiogenesis	Pituitary Tumour Transforming Gene 1 (*PTTG1*)	[112]
miR-148a and miR-31	Up	Promotes angiogenesis	factor inhibiting hypoxia-inducible factor 1 (*FIH1*), and *HIF1α* and *Notch*.	[113]

**Table 2 ijms-21-05631-t002:** The most recent molecules adopted in clinical and preclinical scenarios to treat GBM.

Molecule	Cellular/Molecular Target	Clinical Trial in Glioma	References
Bevacizumab *****	VEGF	Phase III	[5,155,156,157,158]
Lomustine *****	Amino acid carbonylation	Phase III	[131,132]
Nimotuzumab *****	EGFRvIII	Phase II	[135,136]
Veliparib/ABT-888 *****	PARP	Phase II/III	[137,138,139,140,141,159]
Olaparib *****	PARP	Phase I	[142,160]
DNX-2401/Delta-24-RGD *****	Rb pathway-defective glioma cells	Phase I/II	[144,145]
Pembrolizumab *****	PD-1	Phase I/II	[146,147]
MSC-IFN-β *****	MGMT via p53 gene induction		[148]
Momelotinib/MTB/CYT387 *****	JAK 1/2		[149]
Metformin *****	AMPK, p53, mTORC1, mtComplex I	Phase I/II	[150,151,152,153,154]
Bevacizumab and Lomustine	VEGF and Amino acid carbonylation	Phase III	[130]
Lomustine and Regorafenib	Amino acid carbonylation and angiogenic, stromal, and oncogenic receptor tyrosine kinases	Phase II	[133]
Nivolumab	PD-1/PD-L1	Phase II/III	[161,162,163]
DS-100b	IDH1	Phase I	[164,165,166]
Bevacizumab and Pembrolizumab	VEGF and PD-1	Phase II	[167,168]
Anti-TIM-3 and Anti-PD-1	TIM-3 and PD-1		[169]
Anti-TIGIT and anti-PD-1	TIGIT and PD-1		[170]

***** Combination with Temozolomide.

**Table 3 ijms-21-05631-t003:** UP- or DOWN-regulated miRNAs involved in chemoresistant GBM are listed, with their target mRNAs or processes.

miRNA	Target mRNA or Process	In Chemo-Resistant GBM	References
miR-1238	CAV1/EGFR pathway	miRNA is UPregulated	[172]
miR-151 and miR-151a	XRCC4-mediated DNA repair	miRNA is UPregulated	[173]
miR-1587	*NCOR1* in glioma-associated mesenchymal stem cells	miRNA is UPregulated	[82,174]
miR-29a	cancer stem cells	miRNA is DOWNregulated	[176]
miR-29c	Sp1, MGMT	miRNA is DOWNregulated	[177]
miR-423-5p	ING-4, that controls p-AKT and p-ERK1/2; sustain stemness	miRNA is UPregulated	[178]
miR-123-3p	SOX2, WNT/beta-Cat pathway	miRNA is DOWNregulated	[180]
Let-7i, miR-151-3p, and miR-93	unknown	miRNAs are DOWNregulated	[179]
miR-146b-5p	TRAF6, AKT/NF-κB pathway	miRNA is DOWNregulated	[181,184]
miR-223	PAX, PI3K/Akt pathway	miRNA is UPregulated	[185,186]
miR-93; miR-193	Cyclin D1 and cell cycle progression	Both miRNA are UPregulated	[187]
miR-7-5p; miR-186	YY1, in the RelB pathway	Both miRNA are DOWNregulated	[188,189,190]
miR-224-3p	HIF-1α/miR-224-3p/ATG5 pathway	miRNA is UPregulated	[191]
miR-519a	STAT3/Bcl-2/Beclin-1 pathway	miRNA is DOWNregulated	[192]
miR-138	miR-138/BIM axis and autophagy	miRNA is UPregulated	[193]
miR-20a	Under the control of DNMT	miRNA is UPregulated	[194]
miR-139	GFAP in cytoskeleton	miRNA is DOWNregulated	[195,196]
miR-221/222	APE1/miR-221/222 axis regulates SIRT1/MDR1 and PTEN	miRNA is DOWNregulated	[197,198,200,201]
miR-191	Controlled by ER alpha and beta	miRNA is DOWNregulated	[206]
miR-101	GSK3β	miRNA is DOWNregulated	[207]
seven miRNAs’ profile: miR-1280, miR-1238, miR-938, and miR-423-5p; let-7i, miR-151-3p, and miR-93		miRNAs are Upregulated miRNAs are DOWNregulated	[179]
seven miRNAs’ profile: miR-9, -182, and -221; miR-29c, -93, -101, and -130a	miR-101 controls GSK3β	miRNAs are Upregulated; miRNAs are DOWNregulated	[210]

**Table 4 ijms-21-05631-t004:** PET radiotracers for CNS tumours and tumour microenvironment.

Imaging Target	Molecular Target	Radiotracer	References
Glucose metabolism	GLUT-1; HK-1	[^18^F]FDG	[217]
Glutamine metabolism	ASCT2	[^18^F]F-Gln	[218,219,220]
Fatty acid synthesis	MCT; ACSS2	[^11^C]Acetate	[221,222]
Tryptophan pathway/immune tolerance	IDO1	[^11^C]AMT; L-1-[^18^F]FETrp	[212,223,224,225,226,227]
Aminoacids	LAT1/2	[^11^C]MET; [^18^F]FET; [^18^F]FELP; [^18^F]FIMP; [^18^F]Fluciclovine	[228,229,230]
Cell proliferation	TK1	[^18^F]FLT	[231,232,233,234,235,236]
Choline metabolism	CK	[^11^C]Choline	[237,238]
Vasculature and hypoxia	Tumor hypoxia	[^64^Cu]ATSM; nitroimidazole compounds; CAIX agents	[239,240,241,242,243,244]
	α_V_β_3_ ligand-binding domain	[^64^Cu]RaftRGD	[242]
Glioma associated macrophages and microglia (GAMMs)	TSPO	[^11^C](R)PK11195; [^18^F]DPA-714; [^18^F]GE-180; [^18^F]VC701	[245,246,247,248,249,250,251,252,253]
Endothelial cells/astrocytes	PSMA	[^68^Ga]PSMA; [^18^F]DCFPyL; [^89^Zr]Df-IAB2M anti-PSMA minibody	[254,255,256,257,258,259,260,261,262,263,264,265,266]
Matrix-metallo proteinases	MT1-MMP (MMP-14); MMP-2; MMP-9	[^89^Zr]DFO-LEM2/15; [^18^F]BR-351; [^18^F]iCREKA; [^18^F]-FP-chlorotoxin; [^68^Ga]/[^64^Cu]-MMP-14 binding peptide	[246,267,268,269,270]
Carcinoma-associated fibroblast	FAP	[^117^Lu]/[^68^Ga]FAPI-02/04	[271]

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
