# Peer review of "Molecular and Cellular Complexity of Glioma. Focus on Tumour Microenvironment and the Use of Molecular and Imaging Biomarkers to Overcome Treatment Resistance"

_ijms, 2020, doi:10.3390/ijms21165631_

Round 1
Reviewer 1 Report
The manuscript by Valtorta et al. represents long comprehensive review which aims to highlight “… the importance and the complexity of tumour microenvironment in progression and in therapy resistance of glioma.” In common, this is a well-written review attempting to summarise different aspects of GBM development and treatment, and describe in details selected molecular components involved in this process. It combines sections devoted to specific gene mutations, the possible functions of several non-coding microRNAs and the heterogeneity of cell types (sections 2 and 3, 15 pages of 26, 211 references of 338) and the only section on molecular imaging of GBM heterogeneity and microenvironment (section 4) reflected in the manuscript title. As the result, the title of the manuscript does not correspond completely to the overall information presented in the text and may mislead the reader about the content of this review.
To my mind, it would be reasonable either correct the title and abstract (in part) of this review (to reflect all the sections of this manuscript as a whole) or split the manuscript to 2 reviews with more targeted research topics and readership.
Minor points
Key words – please, remove excessive dashes.
Table 4 legend – “microenvironment” instead of “micro-environment”.
Author Response
Comments and Suggestions for Authors
The manuscript by Valtorta et al. represents long comprehensive review which aims to highlight “… the importance and the complexity of tumour microenvironment in progression and in therapy resistance of glioma.” In common, this is a well-written review attempting to summarise different aspects of GBM development and treatment, and describe in details selected molecular components involved in this process. It combines sections devoted to specific gene mutations, the possible functions of several non-coding microRNAs and the heterogeneity of cell types (sections 2 and 3, 15 pages of 26, 211 references of 338) and the only section on molecular imaging of GBM heterogeneity and microenvironment (section 4) reflected in the manuscript title. As the result, the title of the manuscript does not correspond completely to the overall information presented in the text and may mislead the reader about the content of this review.
To my mind, it would be reasonable either correct the title and abstract (in part) of this review (to reflect all the sections of this manuscript as a whole) or split the manuscript to 2 reviews with more targeted research topics and readership.
We thank the reviewer for this observation. We changed the title of the manuscript into “Molecular and cellular complexity of glioma. Focus on tumor microenvironment and the use of molecular and imaging biomarkers to overcome treatment resistance“ (highlighted in yellow in the Revised manuscript). In relation to the new title, we also modify the Abstract accordingly (changes in yellow).
Minor points
Key words – please, remove excessive dashes.
We thank the reviewer for this observation. We removed excessive dashes (highlighted in yellow in Keywords section)..
Table 4 legend – “microenvironment” instead of “micro-environment”.
We thank the reviewer for this observation. We corrected the typing error (highlighted in yellow in Table 4 legend).
Reviewer 2 Report
This is a comprehensive review article describing current and past research and current and future applications regarding the role of imaging modalities investigating tumor microenvironment potentially to evaluate tumor resistance in gliomas. It is well organized by category. It accurately summarizes extensive investigations in this subject matter and accurately reflects the references. I found this to be an informative synthesis of the current state-of-the-art.
In section 5, I did not get the sense that future direction of the described technologies (i.e. their potential clinical applications) are fully described and would like to see this expanded upon.
Author Response
Comments and Suggestions for Authors
This is a comprehensive review article describing current and past research and current and future applications regarding the role of imaging modalities investigating tumor microenvironment potentially to evaluate tumor resistance in gliomas. It is well organized by category. It accurately summarizes extensive investigations in this subject matter and accurately reflects the references. I found this to be an informative synthesis of the current state-of-the-art.
In section 5, I did not get the sense that future direction of the described technologies (i.e. their potential clinical applications) are fully described and would like to see this expanded upon.
We thank the reviewer for this observation. We modified the Section 5 “Final Considerations and future direction” (novel part in green).